# Influence of Rehabilitation Aid Use on Obstacle Height During Gait in Patients with Foot Drop: A Case Series Study

**DOI:** 10.3390/healthcare13222984

**Published:** 2025-11-20

**Authors:** Joonsung Park, Himchan Shim, Changho Jang, Hanyang Yin, Jongbin Kim

**Affiliations:** 1Division of Kinesiology, Silla University, Busan 46958, Republic of Korea; joon3750@silla.ac.kr (J.P.);; 2Graduate School of Education, Silla University, Busan 46958, Republic of Korea

**Keywords:** stroke, foot drop, obstacle crossing, gait, biomechanics

## Abstract

**Highlights:**

**What are the main findings?**
In patients with foot drop, assisted gait (AG) improved dorsiflexion and frontal plane stabilization, as well as knee flexion and frontal plane alignment, thereby contributing to toe clearance and initial contact stabilization.The use of rehabilitation aids improved mechanisms for initial contact and propulsion transition during obstacle crossing, thereby contributing to shock absorption and enhanced dynamic stability.

**What is the implication of the main finding?**
As the required clearance angle increases with obstacle height, AG is considered advantageous for safe walking in everyday environments with uneven surfaces, curbs, and thresholds.Rehabilitation aids provide support for patients with foot drop, with reduced toe tripping and improved gait safety.

**Abstract:**

Background/Objectives: The study explored differences in gait characteristics and biomechanics according to rehabilitation aid use (normal gait (NG) vs. assisted gait (AG) groups) and obstacle height (0, 5, and 15 cm conditions) in patients with stroke-induced foot drop. Methods: A longitudinal study, within-subjects, repeated-measures study was conducted in 10 patients with mild foot drop. Participants walked at their preferred speed on a 10-m indoor track while crossing obstacles of three heights (0, 5, and 15 cm) under two gait conditions (normal walking (NG) and assisted walking (AG). The order of gait conditions and obstacle heights was randomized clinical within participants. Synchronized 3D motion capture with force plate measurements was used to calculate spatiotemporal variables, including ground reaction force (GRF), lower extremity joint angles at heel contact (HC), and toe-off (TO). A two-way repeated-measures ANOVA was used to examine the main effects and interaction effects of gait condition (NG vs. AG) and obstacle height (0, 5, and 15 cm). Results: AG showed little change in gait pattern, while there was a significant interaction between height and group. The vertical GRF (Fz) was lower in AG than NG at 5 cm, indicating reduced initial impact. There was a significant interaction for right medial–lateral (ML) GRF, with AG showing a small ML directional GRF over low obstacles (0–5 cm). At HC, AG was associated with an increase in right ankle dorsiflexion and right knee flexion. AG led to a reduction in left hip angle in the sagittal plane, and a smaller right ankle angle in the frontal plane, suppressing ML sway. At TO, AG caused an increase in right knee flexion, and bilateral ankle angles in the frontal plane at 5 cm. Conclusions: Rehabilitation aids reduced impact at initial contact, enhanced frontal plane stability, improved knee flexion and ankle alignment during the propulsion transition phase, and contributed to reduced variability and improved gait stability. The findings suggest potential utility for public health implications ambulation over curbs and thresholds, warranting larger, adequately powered clinical outcome trials.

## 1. Introduction

Stroke is a representative cerebrovascular event characterized by localized neurological deficits due to abnormal cerebral blood flow; it is among the leading causes of death and disability worldwide [1,2]. After acute stages of injury, neurological deficits such as sensory impairment, ataxia, and hemiplegia are accompanied by muscle weakness and imbalance, impaired postural control, spasticity, reduced voluntary motor control, and abnormal body alignment, leading to poorer balance and gait function due to physical asymmetry and limited functional movement [3,4].

While evidence indicates that approximately 60–80% of patients recover independent walking after rehabilitation [5,6], localized impairments such as foot drop remain a significant functional limitation even after recovery. Foot drop is observed in approximately 20% of patients with stroke [7], and causes abnormal gait patterns such as difficulty achieving heel contact (HC), foot dragging, circumduction, and excessive hip flexion [8]. These changes result in slower gait speed and a higher risk of falls, thereby limiting daily mobility and safety.

Foot drop symptoms can be managed using different techniques and devices, including ankle foot orthoses (AFOs), functional electrical stimulation (FES), physical therapy, surgery, nerve blocks, or medications [9,10,11]. AFOs improve spatiotemporal variables of gait and ankle/knee kinematics, with consistent effects particularly in enhancing toe clearance during the swing phase and preventing tripping [12,13]. FES induces active dorsiflexion (DF) aligned with the gait cycle by stimulating the peroneal nerve/anterior tibial muscle, improving walking speed and functionality, and provides additional benefits when combined with rehabilitation training. However, prolonged FES may cause fatigue and skin-related issues [14,15,16,17]. Despite these device-level benefits, it remains unclear whether assisted walking specifically alters the joint angles that determine heel contact and foot clearance during toe-off when navigating obstacles. This is a common real-world problem in foot-off situations. Therefore, determine whether assisted walking alters vertical impact and frontal control at critical moments across obstacle heights.

Recently, approaches focusing on shoe structure, material stiffness, fixation methods, insole design, shoe–orthosis interactions, and foot-switch-based FES triggering have been developed to optimize gait phase synchronization and mechanical assistance during the propulsion phase [18,19,20,21]. To our knowledge, there are presently no shoes that provide immediate electrical stimulation triggered by in-shoe insole signals during walking, and dedicated rehabilitative footwear designed specifically for patients with foot drop is not yet available.

Pedestrian routes are characterized by straight and curved paths, slopes, stairs, moving crowds, and fixed/variable obstacles during physical activity [22]. Patients with foot drop often experience limitations in gait function under unexpected environmental changes, such as sudden variations in obstacle height or position, or interference from moving crowds [23]. In particular, obstacle crossing requires simultaneous control in horizontal and vertical directions, anticipatory postural adjustments, and coordination of the lower limbs [24,25]. With increased visual dependence when focusing on obstacles [26], the support side (affected side) becomes unstable, resulting in worsened instability, reflected through indicators such as a shortened swing phase, reduced speed, and extended swings [27,28]. A recent study focused on older adults also found that gait performance declined during tasks requiring obstacle crossing while looking around [29]. Biomechanically, obstacle clearance increases vertical toe-off demands and shifts the center of mass trajectory, requiring anticipatory postural control and precise frontal foot alignment. Neuromotor control relies on visual-vestibular integration and feedforward planning. In poststroke foot drop, decreased dorsiflexion and altered hip-knee strategy increase the risk of toe-off interference and lateral instability. Therefore, this study should focus on analyzing the peak vertical ground reaction force (GRF) at initial contact and the sagittal and frontal ankle/knee angles at heel contact and toe-off.

Obstacle-crossing tasks are often used for evaluating functional mobility and validating intervention effects in patients with foot drop, with relatively rich comparative evidence for AFOs and FES. However, there have been limited simultaneous explorations of gait patterns, kinematics, dynamic balance, and safety under varying obstacle heights using functional shoes and insoles that provide electrical stimulation. The purpose of ambulatory chronic stroke (≥6 months) with spasticity level ≥ 3 on the Modified Ashworth Scale (MAS) foot drop to minimize spontaneous recovery effects and focus on device-responsive biomechanical mechanisms. Addressing this knowledge gap, the current study quantitatively evaluated spatiotem-poral/kinematic parameters and dynamic balance at different obstacle heights in patients with stroke-induced foot drop, both with and without rehabilitation aids, aiming to pro-vide fundamental evidence-based data necessary for standardizing clinical prescriptions and training protocols.

## 2. Materials and Methods

### 2.1. Study Design

Prospective single-arm case series with a within-subject repeated-measures design comparing normal gait (NG) and assisted gait (AG) across three obstacle heights (0, 5, 15 cm). Each participant served as their own control; three trials per condition with enforced rests were performed on a 10 m indoor track. The order of NG/AG blocks and obstacle heights was randomized and counterbalanced where feasible, with familiarization passes provided to mitigate learning effects. Given the case-series nature and device/safety constraints, the sample size was pragmatic (*n* = 10).

### 2.2. Participants

This study involved 10 patients with stroke-related mild foot drop living in City B (Table 1). Patients who had received a stroke diagnosis at least six months prior; were able to walk independently; had no orthopedic surgery or fractures within the past six months; exhibited no history of cardiovascular instability; and had a cognitive function score < 25 on the Korean version of the Mini-Mental State Examination (K-MMSE) and spasticity level ≥ 3 on the Modified Ashworth Scale (MAS), with the right leg as the affected side were included in this study. Approval was obtained from the Institutional Review Board (IRB) of the authors’ affiliated institution before participant engagement (1041449-202505-HR-001). Participants were given a detailed explanation of the study and the possible risks and they provided informed consent (by signing a consent form) prior to participation. To reduce between-subject heterogeneity in hemiparetic patterns, we restricted inclusion to right-side involvement. All measurements were conducted under identical conditions, which improves internal validity for within-subject comparisons but restricts generalizability to left-sided lesions.

### 2.3. Measurement Procedure

For measurements, the gait conditions were divided into normal gait (NG) and assisted gait (AG). For NG, the participants were wearing their regular shoes. For AG, they wore functional shoes for foot drop, an electrical stimulation device, and an assistive device. The obstacles consisted of a rigid synthetic resin surface with a width of 60 cm and heights of 0, 5, and 15 cm. The participants walked at their preferred speed, starting from a 3-m approach line [30]. Each condition was repeated three times on an indoor 10 m track, with 1–2 min of rest between attempts. Failure criteria (contacting/stepping over/stopping at obstacles) were set beforehand, and a safety officer assisted participants nearby.

Before the measurements, a body composition analyzer (Inbody, Seoul, Korea) was used to quantitatively measure participants’ height, body weight, skeletal muscle mass, and body fat mass. We requested participants to limit caffeine and alcohol intake and adjust meal times on the day before the measurements. During measurements, participants were instructed to stand with their arms slightly spread apart at an approximately 15° angle. Subsequently, participants’ cognitive function was assessed using the Korean Version of Mini-Mental State Exam K-MMSE (score range: 0–30 points). Following standardized guidelines, the test was conducted in a quiet environment through one-on-one interviews, generating a total score and scores for each domain (orientation, memory registration/recall, attention/calculation, language, and spatiotemporal organization). The total score was interpreted considering the influences of age and education level. The screening criteria were scores <24 points (suspected dementia) based on previous studies, or <26 points (suspected mild cognitive impairment with sensitivity prioritized) based on the study objective. To ensure reliability, examiners received training in advance. No repeat testing was performed on the same participants.

For gait analysis, camera calibration was performed using T-wand (Qualisys, Sweden) to establish three-dimensional spatial coordinates. Coordinates were defined as follows: left-right direction on the X-axis, anterior–posterior direction on the Y-axis, and vertical direction relative to the ground on the Z-axis. After the participants put on lower body stockings, a total of 35 markers were attached to the lower limb segments and joint points. To model the lower limb into seven segments, we placed markers on the following areas, as shown in Figure 1: Bilateral iliac crests, anterior superior iliac spines, posterior superior iliac spines, coccyx, greater trochanter, front thigh, medial/lateral knee, front shank top/bottom, medial and lateral shank, medial and lateral malleolus, first/fifth metatarsal head, second distal phalanx, and calcaneus [31].

The test was conducted three times for each combination of gait condition (NG and AG) and obstacle heights (0 cm, 5 cm, and 15 cm) was randomized within participants. Participants walked at their preferred speed on a 10-m indoor track (Figure 2). They were given time for ample rest before and after each walking attempt to minimize the accumulation of fatigue. In the AG condition, an electrical stimulation band device (Elmusband, Coremovement, Republic of Korea) was attached to the anterior tibial muscle (common peroneal nerve region) of the participants’ affected side, with the frequency set to 35 Hz and the pulse width to 350 μs [32]. Stimulation intensity was individually adjusted to approximately 60% of the maximal tolerable intensity perceived by each participant. The pressure sensor insoles and the electrical stimulation device used in this study were designed to be synchronized in real time, such that the electrical stimulation could be automatically controlled according to the gait phase. The swing phase was detected based on the transition point where pressure on the forefoot of the unaffected side shifted from 0 to positive (ground contact loss, toe-off (TO)). At this point, electrical stimulation was delivered to the anterior tibial muscle of the affected side (35 Hz, 350 μs, individualized intensity: ~60% of perceived maximum) to induce DF. A foot drop shoe was connected to the electrical stimulation device, and an assistive device was also used.

### 2.4. Measurement Devices

The participants walked while wearing an electrical stimulation device, foot drop shoes, assistive devices, and pressure sensor insoles (Figure 3). The characteristics of the devices used under the AG condition are as follows. For the foot drop shoes, a Velcro tension system was applied for easy attachment and removal, and the internal space around the forefoot and instep was increased to provide participants with a comfortable fit. It weighed approximately 426 g for a shoe size of 270 mm. The outer material and the tongue were made of synthetic leather and mesh fabric, respectively, ensuring both breathability and durability.

The assistive device for the lower limb was made of acrylonitrile butadiene styrene, making it highly shock-absorbent and durable. It was made with a height of approximately 38.1–47.3 cm, a width of 23.2 cm, and a depth of 13 cm, allowing the height to be adjusted to fit the lower leg length of each participant. The angle between the legs of the assistive device was set to 75° to increase shock absorption capacity. It was designed to cover the upper part of the shoe, maximizing the fit with the shoe.

The electrical stimulation device was a compact wearable one with dimensions of 54.2 × 52.2 × 15.3 mm and a weight of approximately 120 g, made from neoprene material for easy wearability and equipped with an on/off button, Bluetooth functionality, and C-type charging. The insoles that can operate the pressure-sensing electrical stimulation device were embedded with a 4-channel film-type pressure sensor based on a flexible printed circuit board. Its module was 2.5 × 2.5 cm in size, and powered by a 3.7 V lithium battery, which can be used for approximately 72 h when fully charged.

### 2.5. Data Collection and Processing Methods

Kinematics data of the lower limb joints during walking were collected using six Qualisys Miqus M3 (Qualisys, Gothenburg, Sweden) infrared cameras at 100 Hz. The three-dimensional position coordinates based on reflective markers were labeled using Qualisys Track Manager (Qualisys, ver 2024.2, Sweden) software. For each participant, gait data were collected through three repeated measurements, converted into a C3D file format, and then preprocessed using a Butterworth low-pass filter with a cut-off frequency of 6 Hz to remove high-frequency noise. Data preprocessing and refinement were performed using Python-based scripts (ver 2.12.12), employing open-source libraries such as NumPy (ver 2.0.2), Pandas (ver 2.0.2), SciPy (ver 1.16.3), and Matplotlib (ver 3.10.0).

### 2.6. Definitions of Gait Analysis Intervals and Events

To obtain consistent gait patterns for participants, we analyzed data from one-step intervals measured on a force plate. The gait cycle was defined as the point when the heel of the same foot first touched the ground (HC) until that heel touched the ground again. Within this cycle, we defined the stance phase as the interval from HC to TO, and the swing phase as the interval from TO to heel strike (HS) (Figure 4).

### 2.7. Definition of Joint Angles of the Lower Limb

Joint angles of the lower limb were calculated based on the joint coordinate system (JCS), applying the three-dimensional rotation definition method. The angle calculation sequence followed the default settings of the Qualisys Track Manager using the standing anatomical position as the reference. All joint angles were presented in degrees (°). The global coordinate system defined the X-axis as left with (+X), Y-axis as forward with (+Y), and Z-axis as upward with (+Z), which were also used to set the direction of the ground reaction force (GRF).

The positive and negative directions for each joint were defined as follows: In the sagittal plane, knee flexion was defined as (+) and extension as (−); hip flexion was defined as (+) and extension as (−). For the ankle joint, DF was defined as (+) and plantarflexion as (−). In the frontal plane, knee joint adduction was defined as (+) and abduction as (−). For the hip joint, adduction was defined as (−) and abduction as (+). For the ankle joint, inversion was defined as (+) and eversion as (−). Finally, in the transverse plane, internal rotation was defined as (+) and external rotation as (−) for all knee, hip, and ankle joints (Figure 5).

### 2.8. Statistical Processing

The mean and standard deviation of the measured values in this study were calculated using the SPSS Ver. 25 statistical software for Windows (SPSS, Inc., Chicago, IL, USA). Two-way repeated measures ANOVA was conducted to determine the differences and effects on spatiotemporal/kinematics and dynamic balance depending on the obstacle height and presence or absence of rehabilitation aids. The Bonferroni test was used for post hoc analysis, with a significance level set at α = 0.05. The effect size for the mean difference between independent samples was calculated using Cohen’s d. For small samples, it was adjusted using Hedges’ g, with effect sizes categorized as small (0.20), medium (0.50), and large (0.80). Sample size was pragmatically limited to n = 10 due to device availability and safety oversight for obstacle tasks. We therefore report a sensitivity analysis (G*Power, α = 0.05, two-way repeated-measures ANOVA with within-subject factors; assumed correlation among repeated measures = 0.50, non-sphericity correction = 0.50), indicating the design was powered to detect at least medium-to-large interaction effects. We treat all inferential results as preliminary and emphasize effect sizes (Hedges’ g).

## 3. Results

### 3.1. Spatiotemporal Gait Variables

There was a statistically significant difference between groups at a step length of 15 cm (*p* < 0.05), and a statistically significant interaction between obstacle height and group (F = 3.406, *p* = 0.041). For step time, a statistically significant difference between groups was also observed at 15 cm (*p* < 0.05), with a significant difference between AG and NG (F = 13.094, *p* = 0.001). In cadence, there was a group difference at 0 cm (*p* < 0.05). Effect verification revealed statistical differences in height (F = 4.115, *p* = 0.022) and the interaction between height and group (F = 10.292, *p* = 0.000). Post hoc analysis showed 15 cm > 5 cm. In the right swing phase, post hoc analysis indicated 0 cm > 5 cm, while no other statistically significant differences were observed.

### 3.2. GRF and CoP

There was a significant interaction in horizontal anterior–posterior GRF (Fx, right) (F = 8.438, *p* = 0.001), with AG showing a significant group difference over NG at 0 cm, while NG showed significantly greater GRF at 15 cm. In the vertical GRF (Fz, right), the main effect of group was significant (F = 4.564, *p* = 0.038), with significantly greater values noted for NG than AG at 5 cm. No significant group difference was found in CoP (*p* > 0.50) (Table 2).

### 3.3. Joint Angles of the Lower Limb at HC

For the hip joint, there was no significant effect on the right side in the sagittal plane, whereas a significant main effect between groups was observed on the left side (F = 10.574, *p* = 0.002). At 0 cm, NG exhibited a larger angle than AG (*p* < 0.05). In the frontal plane, there was a significant difference in height on the left side (F = 3.525, *p* = 0.037), with a trend of greater values for the 15 cm condition than the 0 and 5 cm conditions (c > b,a). There was no significant interaction between group and height. In the transverse plane, there was a significant interaction between height and group on the left (F = 4.566, *p* = 0.015). A statistical difference was observed at 5 cm on the left (*p* < 0.05), while no significant difference was found on the right.

For the knee joint, there was a significant main effect of group in the sagittal plane on the right (F = 4.901, *p* = 0.031). AG showed a generally larger flexion than NG at the obstacle height of 5 cm (NG: 10.93, AG: 13.21), while no statistical differences were found for other variables.

For the ankle joint, there was a significant main effect of group for both sides in the sagittal plane (right F = 14.505, *p* < 0.001; left F = 4.15, *p* = 0.047). At 5 cm on the right, AG had a larger angle than NG (NG: 10.84 ± 4.60, AG: 14.43 ± 6.98), with a statistical difference between NG and AG at 0 and 5 cm (*p* < 0.05). In the frontal plane, a significant group main effect was observed on the right side (F = 61.045, *p* < 0.001), with statistical differences between NG and AG at 0, 5, and 15 cm (*p* < 0.05). In the transverse plane, a main effect between groups was statistically significant for both sides (right: F = 19.490, *p* < 0.001; left: F = 11.111, *p* = 0.002). A statistically significant difference was observed between NG and AG at 15 cm on the right and at 0 and 15 cm on the left (*p* < 0.05) (Figure 6) (Table 3).

### 3.4. Joint Angles of the Lower Limb at TO

For the hip joint, there was a significant main effect of height in the transverse plane (left) (F = 5.033, *p* = 0.010; c > a,b), while no differences were observed for other variables. The sagittal plane of the knee (left) exhibited a significant main effect of group (F = 10.559, *p* = 0.002), with statistically significant differences between NG and AG at 5 and 15 cm (*p* < 0.05). The effect sizes were medium and large for each height, respectively.

For the knee joint, there were significant main effects of group in the sagittal plane (right) (F = 10.559, *p* = 0.002) and in the transverse plane (right) (F = 11.007, *p* = 0.002). Moreover, statistical differences between NG and AG were observed at 5 and 15 cm in the right sagittal plane, 0 cm in the left frontal plane, and 0 cm in the right transverse plane (*p* < 0.05).

For the ankle, the frontal plane (bilateral) showed significant differences in the main effect between groups (right F = 7.710, *p* = 0.008; left F = 6.394, *p* = 0.015), while the transverse plane (left) revealed a main effect and an effect size (F = 27.077, *p* < 0.001). Moreover, there were statistical differences between NG and AG at 5 cm in the right frontal plane; 15 cm in the left frontal plane; and 0, 5, and 15 cm in the left transverse plane (*p* < 0.05) (Figure 6) (Table 4).

## 4. Discussion

Prior studies have shown that owing to muscle weakness and limited control of lower limb joints caused by hemiplegia, patients with foot drop tend to exhibit impaired gait performance under obstacle conditions [33,34]; our findings are consistent with these prior results. In terms of gait patterns, NG was associated with an increase in step length as obstacle height rose from 0 to 15 cm (0 cm: 47.50 ± 7.40 cm → 15 cm: 53.00 ± 6.30 cm), whereas AG showed little change (0 cm: 48.80 ± 7.70 cm → 15 cm: 48.50 ± 8.80 cm). Step time was shorter in AG than in NG, while there were some differences in cadence depending on the height condition. This outcome is consistent with evidence indicating a link between wearing rehabilitation aids (AG) and a strategy to minimize fall risk factors by lifting the foot higher (stepping-over) to create space between the toes and obstacles during obstacle crossing, and shortening the landing distance [35]. Furthermore, our results indicate that hemiplegic patients with varying degrees of spasticity tend to maintain dynamic stability by reducing step length, adjusting cadence, and modifying step length in response to increasing obstacle height [36]. Furthermore, the present study found no differences in swing and stance phases between bilateral sides. Considering previous evidence indicating that ensuring stable bilateral step length during obstacle crossing is crucial for stabilization [21,37], our results suggest that variability in spatiotemporal variables become lower even with the use of rehabilitation aids and changes in obstacle height, contributing to improved stability and consistency in gait function.

In GRF analysis, AG was associated with significantly lower vertical component (Fz) peaks than NG at an obstacle height of 5 cm (F = 4.564, *p* = 0.038; NG 2.19 ± 0.65 BW vs. AG 1.92 ± 0.56 BW, approximately −12%). This outcome indicates that AG reduced the initial impact component by inducing a “soft landing” during initial contact, which is consistent with evidence showing that paretic propulsion is closely linked to gait speed and symmetry, and that anterior–posterior GRF-based propulsion parameters explain functional recovery [38,39]. Meanwhile, the result at 15 cm may indicate the selection of stable strategies, such as reduced step length and decreased propulsive peak, to avoid toe-in collisions and ensure support stability. There was no statistical difference in the anterior–posterior component (Fy), suggesting that in AG, anterior–posterior sway might have been managed through localized control of frontal plane alignment without overly increasing medial or lateral deceleration or acceleration. There have been reports that obstacle crossing is accompanied by risks of lateral instability [40] and that support instability and reduced speed in patients with stroke increase fall risk [33]; as such, the reduction in the ankle angle in the frontal plane during HC observed in this study could be interpreted as an indicator supporting improved anterior–posterior stability without an increase in Fy. Accordingly, we recommend an approach that prioritizes securing stability through foot alignment support for patients with unstable gait, while utilizing rehabilitation aids specifically designed for the gait characteristics of each individual. On the right leg, the height × group interaction was significant (F = 8.438, *p* = 0.001), while it was not significant on the left leg. From the measurements, on the right side, AG had a greater medial-lateral (ML)-GRF than NG at 0 cm (1.22 N/kg → 1.41 N/kg) and 5 cm (1.14 N/kg → 1.33 N/kg), whereas they were similar at 15 cm (1.14 N/kg, 1.18 N/kg). This result suggests that for low obstacles, wearing the assistive device (AG) should enhance active control in the ML direction (lateral stabilization), rapidly returning the center of mass (COM) to the support surface through frontal plane alignment of the foot-ankle and adjustment of the foot position. At higher obstacles (15 cm), however, no significant additional ML benefits from wearing the assistive device appeared to be evident in either condition. Further, we found no significant difference on the left side, suggesting that the primary adjustment in ML control likely occurred on the right support limb; this seems to represent an increase in active lateral propulsion/inhibition responses (hip abductors/ankle invertor/evertor action) to compensate for excessive lateral sway. The increased ML-GRF could be interpreted as the active implementation of a stabilization goal rather than increased instability. This deduction is consistent with previous studies reporting that anticipatory postural adjustments and lateral stability during obstacle crossing are correlated with fall risk [28,33,40].

As for the joint angle of the lower limb, in AG, participants appropriately utilized the joint angle and timing of the lower limb during the HC and TO phases in accordance with the obstacle height. In comparison to NG, AG was associated with an increase in the DF of the right ankle in the sagittal plane at 5 cm of obstacle height (10.83 ± 4.60° → 14.42 ± 6.98°, F = 14.505, *p* < 0.001), with a statistically significant difference. The right knee joint also showed a statistically significant difference by height (F = 4.901, *p* = 0.031). The left hip joint in the sagittal plane also exhibited a difference between groups (F = 10.574, *p* = 0.002). This result indicates a gait strategy where upon HC, the hip joint bends too much proximally (toward the body), compensating by bending the knee more to lift the ankle. Importantly, AG had a smaller frontal plane angle in the right ankle joint at HC (5 cm, 14.13 ± 7.54° → 7.07 ± 2.95°, F = 61.045, *p* < 0.001), contributing to the suppression of ankle sway (Table 3). Previous studies indicate that peroneal nerve stimulation induces active DF in the early swing phase, facilitating a smooth gait pattern [14], while using assistive devices modulates ankle angle/moment and knee compensation in the early gait phase [41,42]. In addition, the HC phase is a period where shock absorption and the transition to initial body weight support overlap, where attaining DF and maintaining hip and pelvic alignment are considered important [43]. In the comparison between groups in the sagittal plane of the right ankle and the transverse plane of both ankles, we found distinct ankle angle patterns in AG. These findings are consistent with compensatory strategies commonly observed in patients with foot drop, including increased hip flexion, greater external rotation at the hip and knee, and enhanced external rotation of the foot due to the risk of toe interference caused by reduced DF [9,10]. In addition, the left hip angle in the frontal plane increased with obstacle height, suggesting enhanced isometric and eccentric activity of the gluteus medius, as reflected in pelvic tilt and hip adduction–abduction control [35,37,44]. Overall, the results indicate that these lateral stability strategies become increasingly pronounced as the obstacle height increases. In this study, AG improves ankle-joint control across the sagittal, frontal, and transverse planes and produces more favorable event-timing angles at the knee and hip, suggesting that AG acts as a compensatory strategy to secure toe clearance. In line with this, lower vertical impact at initial contact, reduced frontal-plane ankle-angle variability, and greater knee flexion at toe-off are associated with dynamic stability and toe-clearance mechanics.

At TO, right knee flexion in the sagittal plane increased (27.71 ± 7.66° → 32.75 ± 6.30°, F = 10.559, *p* = 0.002), while AG was associated with a greater increase in the ankle flexion in the frontal plane (bilateral) at 5 cm (right: 4.29 ± 5.93° → 8.54 ± 3.06°, F = 7.710, *p* = 0.008; left: 4.43 ± 7.89° → 7.56 ± 6.71°, F = 6.394, *p* = 0.015), suggesting proper segmental coordination depending on the obstacle height. In the cross-sectional plane, however, AG lead to greater ankle rotation angles (bilateral) than NG at 5 cm at HC (Right: F = 19.490, *p* < 0.001; Left: F = 11.111, *p* = 0.002), whereas NG was associated with a larger angle than AG in the left ankle (5 cm, 11.30 ± 5.28° vs. 7.66 ± 3.80°, F = 27.077, *p* < 0.001). These results suggest a gait pattern where the ankle can rotate slightly inward and outward in the transverse plane at HC (Table 4). TO serves as the starting point for push-off and swing transition, where the timing of plantar flexion (PF)−(DF) at the ankle joint and knee flexion determines toe interference during the swing phase [43,45]. In this study, we found that initial knee flexion and ankle alignment were more favorably adjusted in the AG condition for the sagittal plane of the left knee joint (group effect), the cross-sectional plane of the right knee joint (group effect), the frontal plane of both ankle joints (group effect), and the cross-sectional plane of the left ankle joint (group effect). This outcome is consistent with reports on the use of rehabilitation aids among patients with foot drop, which ensure proper knee flexion and stabilize foot inversion, eversion, and rotation [46,47]. In particular, the results suggest that foot drop shoes effectively control the ankle joint by limiting unwanted movements such as inversion/eversion and rotation and influencing anterior–posterior stability, even with differences in height on the left hip joint in the cross-sectional plane, depending on shoe type and obstacle height. Therefore, the increased ankle DF and knee flexion observed at HC, the reduced frontal plane variability, and the increased knee flexion at TO in this study may have led to a complex effect, such as shock absorption during aligned foot contact and stabilization during gait.

With advances in digital rehabilitation, options for post-stroke foot drop include (i) ankle–foot orthoses (AFOs), (ii) surface functional electrical stimulation (FES), and (iii) robot-assisted or sensor-augmented wearables. AFOs are widely accessible and affordable but do not control event-specific timing during gait [15]. FES can improve dorsiflexion timing/toe clearance; commercial systems are mid-cost and usually require brief training (≈USD 4000–7000) [17,48]. Wearable robots/exosuits are promising yet typically clinic-based and require specialist supervision [49]. Our aid-assisted footwear targets heel-strike and toe-off mechanics with minimal clinic time, supporting everyday obstacle negotiation.

In summary, our findings indicate that AG improved DF and frontal plane stabilization during the HC phase, as well as knee flexion and frontal plane alignment during the TO phase, thereby contributing to toe clearance and initial contact stabilization. As the required clearance angle increases with obstacle height, AG is considered advantageous for safe walking in everyday environments with uneven surfaces, curbs, and thresholds. The use of rehabilitation aids improved mechanisms for initial contact (DF/frontal plane stabilization) and propulsion transition (knee flexion/ankle alignment) during obstacle crossing, thereby contributing to shock absorption and enhanced dynamic stability; this provides support for patients with foot drop, with reduced toe tripping and improved gait safety. All participants had right-sided involvement, and hemispheric lateralization can differentially influence postural and cognitive control; therefore, generalization to left-sided lesions warrants caution and targeted replication.

## 5. Conclusions

In this study, we analyzed differences in gait pattern, GRF, CoP, and joint angles in patients with foot drop with and without rehabilitation aids, depending on obstacle height. The results showed that AG conditions tended to partially restore consistency and stability in gait by suppressing variability in gait pattern even when the obstacle height was varied. However, higher obstacle heights (e.g., 15 cm) activated the use of conservative strategies in both the AG and NG conditions (reduction in step length and propulsion), resulting in stability-prioritized gait control. Furthermore, AG was associated with improved dynamic stability due to reduced vertical GRF peak, leading to soft landing at initial contact; increased ankle DF and knee flexion at HC; and reduced frontal plane sway.

These conclusions explicitly reflect our Highlights: improved dorsiflexion and frontal-plane stabilization at heel contact, increased knee flexion and ankle alignment at toe-off, and indicators consistent with enhanced dynamic stability and safer everyday walking. It also exhibited reduced foot dragging and segmental coordination through increased knee flexion and improved ankle alignment at TO. Despite these notable findings, a study limitation must be acknowledged. Limitations include small sample size, homogenous lesion side (right), absence of healthy controls. Future trials should adopt larger, randomized counterbalanced designs with metabolic/EMG measures and clinical outcomes (falls, community mobility).

## Figures and Tables

**Figure 1 healthcare-13-02984-f001:**
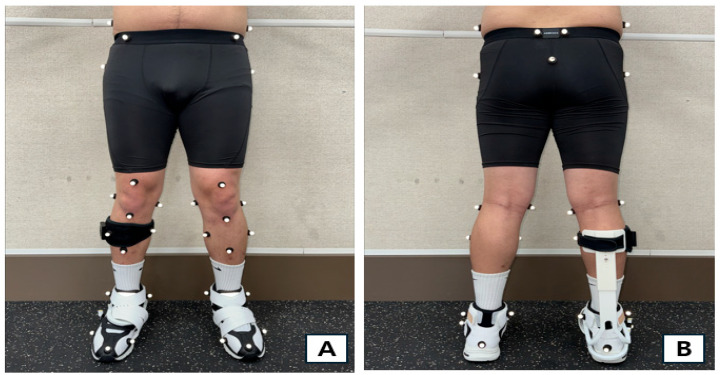
Marker attachment locations for motion analysis. Front (**A**), Rear (**B**).

**Figure 2 healthcare-13-02984-f002:**
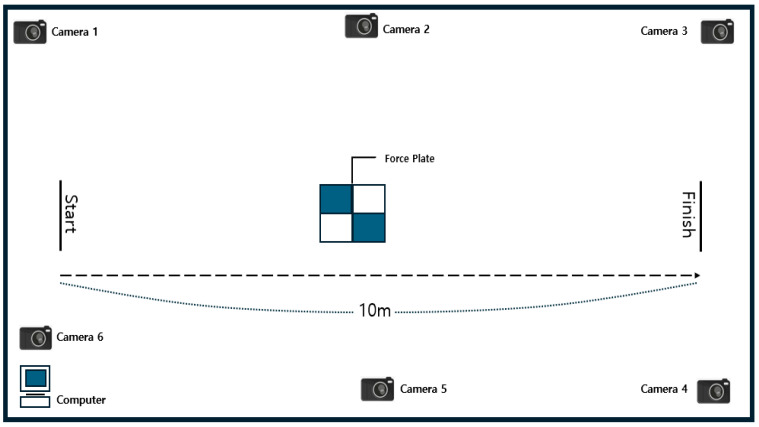
Camera locations and indoor track settings.

**Figure 3 healthcare-13-02984-f003:**
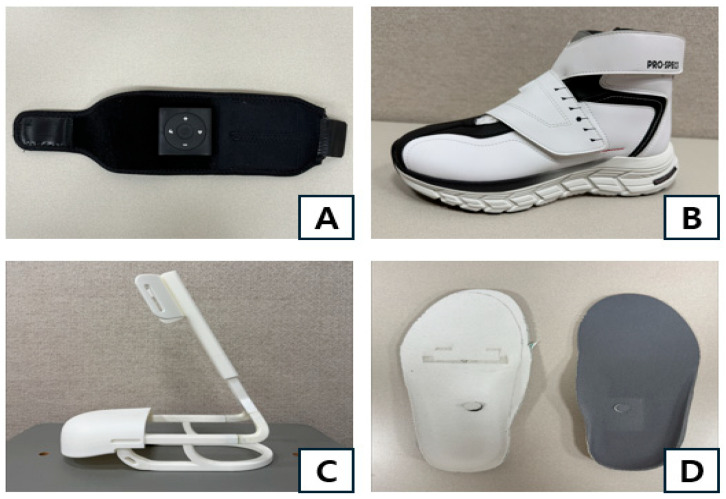
An electrical stimulation device (**A**), functional shoes for foot drop (**B**), an assistive device (**C**), and pressure sensor insoles (**D**).

**Figure 4 healthcare-13-02984-f004:**
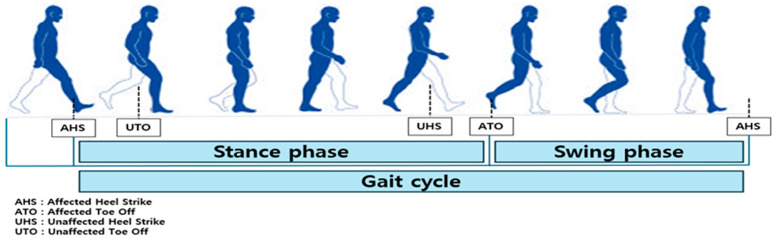
Walking time intervals.

**Figure 5 healthcare-13-02984-f005:**
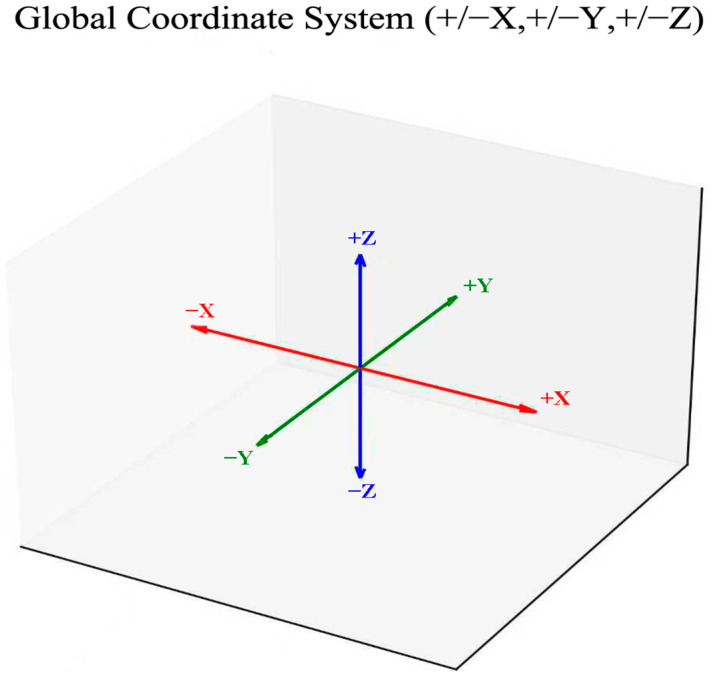
Definition of the joint angle of the lower limb.

**Figure 6 healthcare-13-02984-f006:**
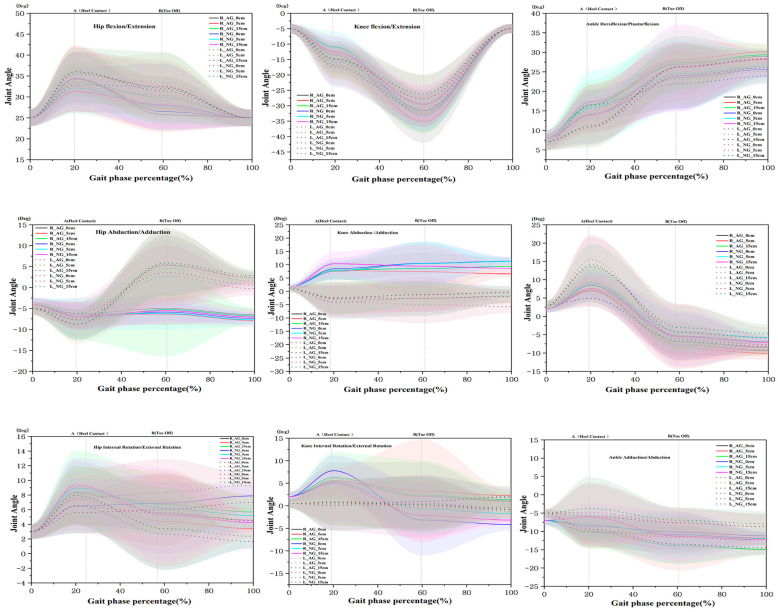
Three-dimensional joint angles of the lower limb by height.

**Table 1 healthcare-13-02984-t001:** Participant characteristics.

No. of People	Age (Year)	Height (m)	Body Weight (kg)	Skeletal Muscle Mass (kg)	Body Fat Mass (kg)	Affected Side
*n* = 10	42.7 ± 2.9	1.61 ± 0.98	58.1 ± 15.5	22.1 ± 7.7	17.8 ± 5.5	Right

**Table 2 healthcare-13-02984-t002:** Comparison of spatiotemporal gait variables, GRF, and CoP between the NG and AG groups by obstacle height.

Parameters		NG	AG	Effect	F (P)	d	Post hoc	Parameters			NG	AG	Effect	F (P)	d	Post hoc
		M(SD)	M(SD)								M(SD)	M(SD)				
Step length (cm)	0 cm **^a^**	47.50 (7.40)	48.80 (7.70)	*H*	0.653 (0.525)	0.310	ns	Medio-lateral GRF (N/kg)	*R*	0 cm **^a^**	**1.22** **(0.34)**	**1.41** **(0.37) †**	*H*	0.247 (0.782)	0.519	*c > a,b*
	5 cm ^b^	48.10 (8.50)	48.40 (6.60)	*G*	1.264 (0.266)	0.081				5 cm ^b^	1.44 (0.51)	1.33 (0.35)	*G*	1.051 (0.310)	0.262	
	15 cm **^c^**	**53.00** **(6.30)**	**48.50** **(8.80) †**	*H*×*G*	**3.406** **(0.0410) ***	1.268				15 cm **^c^**	**1.44** **(0.46)**	**1.18** **(0.32) †**	*H*×*G*	**8.438** **(0.001) *****	0.675	
Step time (s)	0 cm **^a^**	0.63 (0.10)	0.58 (0.15)	*H*	0.494 (0.613)	0.408	ns		*L*	0 cm **^a^**	1.45 (0.61)	1.49 (0.61)	*H*	0.280 (0.757)	0.060	*ns*
	5 cm ^b^	0.60 (0.09)	0.53 (0.14)	*G*	**13.094** **(0.001) ****	0.557				5 cm ^b^	1.45 (0.91)	1.51 (0.65)	*G*	0.029 (0.865)	0.082	
	15 cm **^c^**	**0.64** **(0.16)**	**0.56** **(0.15) †**	*H*×*G*	0.478 (0.623)	0.517				15 cm **^c^**	1.68 (0.60)	1.53 (0.50)	*H*×*G*	0.830 (0.442)	0.277	
Cadence (min/step)	0 cm **^a^**	**97.46** **(11.71)**	**75.80** **(14.04) †**	*H*	**4.115** **(0.022) ***	1.682	c > b	Antero-posterior GRF (N/kg)	*R*	0 cm **^a^**	11.05 (1.10)	11.28 (1.33)	*H*	1.167 (0.320)	0.186	*ns*
	5 cm ^b^	76.23 (13.38)	82.97 (16.86)	*G*	6.174 (0.016) *	0.446				5 cm ^b^	11.80 (1.47)	11.49 (0.88)	*G*	0.144 (0.706)	0.262	
	15 cm **^c^**	91.40 (13.68)	86.94 (13.52)	*H*×*G*	**10.292** **(0.000) *****	0.328				15 cm **^c^**	11.93 (3.36)	11.63 (0.77)	*H*×*G*	0.306 (0.737)	0.145	
Left Swing Phase (s)	0 cm **^a^**	0.58 (0.18)	0.63 (0.21)	*H*	0.224 (0.800)	0.241	ns		*L*	0 cm **^a^**	11.44 (1.51)	11.31 (1.42)	*H*	0.278 (0.759)	0.092	*ns*
	5 cm ^b^	0.65 (0.38)	0.58 (0.15)	*G*	0.901 (0.347)	0.289				5 cm ^b^	11.51 (1.76)	11.44 (1.26)	*G*	0.002 (0.963)	0.045	
	15 cm **^c^**	0.69 (0.22)	0.60 (0.13)	*H*×*G*	1.026 (0.366)	0.508				15 cm **^c^**	11.63 (2.86)	11.81 (1.63)	*H*×*G*	0.192 (0.826)	0.078	
Right Swing Phase (s)	0 cm **^a^**	0.68 (0.18)	0.81 (0.41)	*H*	2.789 (0.071)	0.425	a > b	Vertical GRF (N/kg)	*R*	0 cm **^a^**	1.72 (0.43)	1.81 (0.53)	*H*	2.884 (0.065)	0.194	*c > a*
	5 cm ^b^	0.62 (0.11)	0.60 (0.18)	*G*	0.008 (0.927)	0.096				5 cm ^b^	**2.19** **(0.65)**	**1.92** **(0.56) †**	*G*	**4.564** **(0.038) ***	0.450	
	15 cm **^c^**	0.77 (0.32)	0.64 (0.15)	*H*×*G*	2.231 (0.118)	0.530				15 cm **^c^**	2.37 (0.91)	2.05 (0.62)	*H*×*G*	3.092 (0.054)	0.410	
Left Stance Phase (s)	0 cm **^a^**	0.97 (0.50)	1.17 (0.77)	*H*	0.091 (0.914)	0.309	ns		*L*	0 cm **^a^**	2.11 (0.86)	2.15 (0.83)	*H*	0.705 (0.499)	0.041	*c > a,b*
	5 cm ^b^	1.12 (0.96)	1.18 (0.96)	*G*	0.226 (0.637)	0.060				5 cm ^b^	**2.55** **(1.31)**	**2.09** **(0.91) †**	*G*	2.172 (0.147)	0.413	
	15 cm **^c^**	1.26 (1.15)	1.09 (0.60)	*H*×*G*	1.899 (0.160)	0.191				15 cm **^c^**	2.56 (1.37)	2.51 (1.11)	*H*×*G*	2.122 (0.130)	0.035	
Right Stance Phase (s)	0 cm **^a^**	1.06 (0.54)	0.98 (0.36)	*H*	0.681 (0.511)	0.172		*CoP_X (mm)*			68.43 (24.14)	72.19 (22.36)		0.090 (0.736)	0.162	
	5 cm ^b^	1.33 (1.44)	1.30 (1.17)	*G*	0.016 (0.899)	0.027		*CoP_Y (mm)*			47.50 (21.12)	53.65 (17.00)		0.736 (0.506)	0.323	
	15 cm **^c^**	1.12 (0.95)	1.30 (1.24)	*H*×*G*	0.220 (0.803)	0.163										

Note: ground reaction force, GRF; center of pressure, CoP; normal gait, NG; assisted gait, AG; ^a^, 0 cm high; ^b^, 5 cm high; ^c^, 15 cm high; H, high; G, group; H×G, high × group; * *p* < 0.05; ** *p* < 0.1; *** *p* < 0.001—two-way repeated measures ANOVA; **†**
*p* < 0.05 independent samples *t*-test; d, effect size (Cohen’s d); ns, no significant interaction; R, right; L, left. Bold: Values in bold indicate statistically significant differences (*p* < 0.05).

**Table 3 healthcare-13-02984-t003:** Comparison of heel contact angles between the NG and AG groups by obstacle height.

Variables (Heel Contact)			Hip						Knee						Ankle					
			NG	AG	Effect	F (P)	d	Post hoc	NG	AG	Effect	F (P)	d	Post hoc	NG	AG	Effect	F (P)	d	Post hoc
			M(SD)	M(SD)					M(SD)	M(SD)					M(SD)	M(SD)				
Sagittal (°)	*R*	0 cm **^a^**	34.41 (6.32)	34.36 (7.52)	*H*	0.741 (0.482)	0.008	*ns*	12.49 (4.85)	13.67 (4.07)	*H*	1.439 (0.247)	0.265	*ns*	**10.82** **(4.31)**	**15.48** **(5.99) †**	*H*	0.581 (0.563)	0.905	*ns*
		5 cm ^b^	32.15 (6.23)	34.12 (8.10)	*G*	1.223 (0.274)	0.287		10.93 (5.10)	13.21 (6.24)	*G*	**4.901** **(0.031) ***	0.401		10.83 (4.60)	14.42 (6.98)	*G*	**14.505** **(0.000) *****	0.619	
		15 cm **^c^**	31.27 (4.62)	33.39 (7.05)	*H*×*G*	0.341 (0.713)	0.364		12.55 (2.43)	14.81 (5.64)	*H*×*G*	0.266 (0.768)	0.556		**11.20** **(5.66)**	**16.52** **(8.57) †**	*H*×*G*	0.174 (0.841)	0.747	
	*L*	0 cm **^a^**	**35.51** **(5.30)**	**32.41** **(5.95) †**	*H*	0.216 (0.806)	0.550	*ns*	15.79 (3.14)	16.84 (6.22)	*H*	0.081 (0.922)	0.224	*ns*	14.01 (4.11)	17.30 (3.28)	*H*	0.285 (0.753)	0.891	*ns*
		5 cm ^b^	36.06 (4.87)	33.65 (4.80)	*G*	**10.574** **(0.002) ***	0.496		14.96 (5.98)	16.69 (4.75)	*G*	2.900 (0.095)	0.322		15.98 (5.28)	16.63 (5.37)	*G*	**4.15** **(0.047) ***	0.088	
		15 cm **^c^**	36.00 (5.71)	32.84 (4.21)	*H*×*G*	0.063 (0.939)	0.636		14.44 (4.66)	17.49 (6.35)	*H*×*G*	0.189 (0.828)	0.555		13.89 (4.52)	16.50 (6.47)	*H*×*G*	0.562 (0.573)	0.475	
Frontal (°)	*R*	0 cm **^a^**	7.07 (2.91)	6.25 (2.79)	*H*	0.685 (0.509)	0.288	*ns*	8.47 (3.97)	7.50 (3.48)	*H*	1.309 (0.279)	0.262	*ns*	**15.48** **(4.78)**	**9.45** **(4.32) †**	*H*	2.556 (0.088)	1.086	*a > c*
		5 cm ^b^	6.28 (3.69)	6.536 (2.53)	*G*	0.126 (0.724)	0.081		7.88 (3.25)	7.85 (3.18)	*G*	2.828 (0.099)	0.010		**14.13** **(7.54)**	**7.07** **(2.95) †**	*G*	**61.045** **(0.000) *****	1.345	
		15 cm **^c^**	6.88 (3.33)	7.82 (4.61)	*H*×*G*	1.204 (0.367)	0.247		10.37 (4.55)	8.01 (3.41)	*H*×*G*	0.949 (0.394)	0.592		**13.53** **(6.24)**	**4.92** **(2.15) †**	*H*×*G*	0.644 (0.529)	2.050	
	*L*	0 cm **^a^**	6.16 (3.59)	7.60 (4.12)	*H*	**3.525** **(0.037) ***	0.374	*c > b,a*	2.62 (1.62)	3.94 (1.84)	*H*	0.040 (0.961)	0.231	*ns*	8.53 (3.86)	7.21 (3.29)	*H*	0.076 (0.927)	0.369	*ns*
		5 cm ^b^	7.67 (2.22)	9.65 (3.73)	*G*	2.388 (0.129)	0.666		2.84 (1.08)	2.96 (1.40)	*G*	2.785 (0.101)	0.019		9.09 (3.95)	6.89 (4.25)	*G*	0.961 (0.332)	0.535	
		15 cm **^c^**	8.72 (3.63)	8.70 (3.68)	*H*×*G*	0.661 (0.521)	0.008		2.34 (1.07)	4.07 (1.35)	*H*×*G*	0.670 (0.516)	0.280		7.63 (3.76)	8.43 (4.82)	*H*×*G*	1.640 (0.204)	0.186	
Transverse (°)	*R*	0 cm **^a^**	6.35 (3.76)	8.32 (4.22)	*H*	0.669 (0.517)	0.492	*ns*	7.78 (3.40)	5.57 (3.07)	*H*	1.217 (0.305)	0.590	*ns*	10.15 (4.15)	8.54 (2.92)	*H*	0.375 (0.689)	0.454	*ns*
		5 cm ^b^	8.96 (4.14)	7.46 (3.25)	*G*	0.223 (0.639)	0.407		5.13 (2.11)	5.40 (3.07)	*G*	0.027 (0.869)	0.094		9.88 (3.91)	8.12 (4.75)	*G*	**19.490** **(0.000) *****	0.403	
		15 cm **^c^**	9.36 (3.68)	8.42 (5.62)	*H*×*G*	2.381 (0.103)	0.202		4.95 (2.55)	6.25 (2.62)	*H*×*G*	2.845 (0.068)	0.318		**10.21** **(4.45) †**	**6.04** **(3.68) †**	*H*×*G*	2.726 (0.075)	0.826	
	*L*	0 cm **^a^**	6.46 (3.24)	7.56 (3.81)	*H*	0.078 (0.925)	0.313	*ns*	0.95 (0.58)	0.26 (0.85)	*H*	0.003 (0.997)	0.133	*ns*	**10.08** **(2.67)**	**4.74** **(2.23) †**	*H*	0.232 (0.978)	0.751	*ns*
		5 cm ^b^	**5.59** **(2.77)**	**8.02** **(3.92) †**	*G*	1.645 (0.206)	0.726		0.67 (0.90)	0.59 (0.82)	*G*	0.401 (0.529)	0.003		9.30 (3.93)	8.81 (4.85)	*G*	**11.111** **(0.002) ****	0.545	
		15 cm **^c^**	7.01 (4.03)	6.55 (2.56)	*H*×*G*	**4.566** **(0.015) ***	0.139		0.75 (0.12)	0.01 (0.50)	*H*×*G*	0.101 (0.905)	0.139		**9.40** **(4.34)**	**3.74** **(4.61) †**	*H*×*G*	0.252 (0.778)	0.873	

Note: normal gait, NG; assisted gait, AG; ^a^, 0 cm high; ^b^, 5 cm high; ^c^, 15 cm high; H, high; G, group; H×G, high × group; * *p* < 0.05; ** *p* < 0.1; *** *p* < 0.001—two-way repeated measures ANOVA; **†**
*p* < 0.05 independent samples *t*-test; d, effect size (Cohen’s d); ns, no significant interaction; R, right; L, left. Bold: Values in bold indicate statistically significant differences (*p* < 0.05).

**Table 4 healthcare-13-02984-t004:** Comparison of toe-off angles between the NG and AG groups by obstacle height.

Variables (Toe Off)			Hip						Knee						Ankle					
			NG	AG	Effect	F (P)	d	Post hoc	NG	AG	Effect	F (P)	d	Post hoc	NG	AG	Effect	F (P)	d	Post hoc
			M(SD)	M(SD)					M(SD)	M(SD)					M(SD)	M(SD)				
Sagittal (°)	*R*	0 cm **^a^**	26.49 (4.51)	25.73 (3.58)	*H*	2.457 (0.096)	0.188	*c > a*	29.45 (11.06)	30.85 (5.96)	*H*	0.511 (0.603)	0.164	*ns*	23.59 (8.68)	26.94 (7.25)	*H*	0.311 (0.734)	0.420	*ns*
		5 cm ^b^	27.78 (4.11)	27.28 (5.72)	*G*	0.019 (0.890)	0.103		**27.71** **(7.66)**	**32.75** **(6.30) †**	*G*	**10.559** **(0.002) ***	0.722		24.46 (8.95)	28.12 (4.75)	*G*	2.517 (0.119)	0.535	
		15 cm **^c^**	28.10 (5.56)	29.85 (4.83)	*H*×*G*	0.814 (0.449)	0.337		**26.13** **(6.23)**	**31.59** **(5.29) †**	*H*×*G*	1.130 (0.331)	0.946		26.14 (11.03)	27.34 (8.51)	*H*×*G*	0.202 (0.818)	0.123	
	*L*	0 cm **^a^**	32.40 (5.39)	30.34 (4.61)	*H*	0.350 (0.706)	0.412	*ns*	31.58 (6.63)	35.26 (6.78)	*H*	2.713 (0.124)	0.550	*ns*	23.98 (6.40)	24.05 (8.52)	*H*	0.162 (0.851)	0.009	*ns*
		5 cm ^b^	31.48 (7.32)	33.60 (6.95)	*G*	0.002 (0.963)	0.297		33.24 (5.38)	32.70 (5.71)	*G*	1.954 (0.168)	0.098		26.33 (5.14)	23.14 (6.58)	*G*	3.463 (0.069)	0.544	
		15 cm **^c^**	32.09 (8.65)	32.51 (5.52)	*H*×*G*	0.801 (0.454)	0.059		29.55 (6.55)	32.18 (6.66)	*H*×*G*	0.854 (0.432)	0.398		26.32 (5.88)	21.93 (6.34)	*H*×*G*	0.993 (0.377)	0.717	
Frontal (°)	*R*	0 cm **^a^**	5.87 (2.25)	4.97 (3.84)	*H*	0.206 (0.814)	0.297	*ns*	10.52 (4.16)	8.32 (4.01)	*H*	0.320 (0.728)	0.243	*ns*	5.38 (6.12)	7.84 (2.95)	*H*	0.024 (0.977)	0.543	*ns*
		5 cm ^b^	6.21 (3.85)	6.35 (2.54)	*G*	0.048 (0.828)	0.044		10.42 (6.39)	7.36 (4.94)	*G*	0.648 (0.424)	0.334		**4.29** **(5.93)**	**8.54** **(3.06) †**	*G*	**7.710** **(0.008) ***	0.944	
		15 cm **^c^**	5.54 (3.21)	5.25 (3.19)	*H*×*G*	0.141 (0.869)	0.041		9.45 (4.09)	8.55 (4.66)	*H*×*G*	0.455 (0.637)	0.101		5.63 (8.79)	6.93 (2.50)	*H*×*G*	0.737 (0.484)	0.230	
	*L*	0 cm **^a^**	7.41 (2.74)	7.65 (3.62)	*H*	0.820 (0.446)	0.118	*ns*	**1.15** **(1.42)**	**4.94** **(2.02) †**	*H*	0.155 (0.857)	0.564	*ns*	6.21 (7.56)	7.10 (3.45)	*H*	0.465 (0.631)	0.160	*ns*
		5 cm ^b^	8.18 (4.62)	8.81 (5.43)	*G*	0.717 (0.401)	0.095		5.71 (2.00)	6.75 (4.74)	*G*	3.498 (0.067)	0.007		4.43 (7.89)	7.56 (6.71)	*G*	**6.394** **(0.015) ***	0.428	
		15 cm **^c^**	7.13 (5.29)	7.94 (5.88)	*H*×*G*	0.017 (0.983)	0.078		4.58 (1.30)	6.52 (2.56)	*H*×*G*	1.274 (0.288)	0.227		**3.13** **(2.26)**	**6.81** **(4.10) †**	*H*×*G*	0.712 (0.495)	0.647	
Transverse (°)	*R*	0 cm **^a^**	6.87 (2.83)	5.57 (2.45)	*H*	0.269 (0.765)	0.493	*ns*	**3.21** **(1.90)**	**2.05** **(1.62) †**	*H*	0.405 (0.669)	0.678	*ns*	10.83 (5.10)	10.19 (5.40)	*H*	1.575 (0.217)	0.121	*ns*
		5 cm ^b^	6.21 (3.75)	6.14 (4.39)	*G*	1.141 (0.291)	0.367		0.70 (0.78)	3.24 (1.38)	*G*	**11.007** **(0.002) ***	0.460		13.91 (6.77)	10.96 (6.27)	*G*	3.062 (0.086)	0.452	
		15 cm **^c^**	5.27 (6.37)	6.28 (4.37)	*H*×*G*	1.699 (0.193)	0.187		1.59 (0.41)	2.31 (1.78)	*H*×*G*	0.070 (0.932)	0.593		13.88 (5.09)	11.29 (6.85)	*H*×*G*	0.382 (0.685)	0.434	
	*L*	0 cm **^a^**	**5.49** **(2.56)**	**8.36** **(2.56) †**	*H*	**5.033** **(0.010) ***	0.566	*c > a*,*b*	0.53 (0.11)	0.28 (0.68)	*H*	0.059 (0.943)	0.127	*ns*	**11.64** **(3.31)**	**7.63** **(4.21) †**	*H*	0.731 (0.487)	0.765	*ns*
		5 cm ^b^	5.98 (3.93)	7.10 (3.22)	*G*	1.729 (0.195)	0.243		0.12 (0.47)	1.04 (0.59)	*G*	0.121 (0.730)	0.167		**11.30** **(5.28)**	**7.66** **(3.80) †**	*G*	**27.077** **(0.000) *****	0.800	
		15 cm **^c^**	3.39 (1.64)	2.65 (1.84)	*H*×*G*	2.562 (0.087)	0.141		0.24 (0.11)	0.92 (0.83)	*H*×*G*	0.393 (0.677)	0.115		**13.49** **(5.27)**	**7.16** **(3.44) †**	*H*×*G*	1.291 (0.284)	0.852	

Note: normal gait, NG; assisted gait, AG; ^a^, 0 cm high; ^b^, 5 cm high; ^c^, 15 cm high; H, high; G, group; H×G, high × group; * *p* < 0.05 *** *p* < 0.001—two-way repeated measures ANOVA; **†**
*p* < 0.05 independent samples *t*-test; d, effect size (Cohen’s d); ns, no significant interaction; R, right; L, left. Bold: Values in bold indicate statistically significant differences (*p* < 0.05).

## Data Availability

The raw data supporting the conclusions of this article will be made available by the authors on request.

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
