# Peer review of "Influence of Rehabilitation Aid Use on Obstacle Height During Gait in Patients with Foot Drop: A Case Series Study"

_healthcare, 2025, doi:10.3390/healthcare13222984_

Round 1
Reviewer 1 Report
Comments and Suggestions for Authors
This is a very valuable study.
Please do the following to improve your scientific report.
In the title, replace "A Controlled Experimental Study" with "A Case Series Study ".
In the Methods section, be sure to mention the type of study in the first line and explain your experimental design for this case series study.
Please update your references. 70% of your references should be from the last 5 years.
Given the advancement of digital technology, briefly talk about methods such as those used to correct foot drop and compare it with the method used in your study. From various aspects such as cost, need for an experienced specialist, access to the product and other similar matters. So that you can properly defend your method among the types of rehabilitation tools.
Express your clear message in the Conclusion section.
The items you mentioned in the Highlights section should also be clearly stated in the Conclusion section.
Best regards
Author Response
Dear Reviewer 1,
Thank you for your meticulous review of my paper. Your valuable comments have contributed to its high quality. Your suggestions will enable me to continue to conduct excellent research.
Your comments are summarized in the table below. They are highlighted in red in the manuscript. Thank you again. Have a nice day today.
|
Comment (verbatim) |
Response / Action Taken |
Manuscript Location |
|
Replace "A Controlled Experimental Study" with "A Case Series Study" in the title. |
Thank you, reviewer. Based on your feedback, I changed the title to "A Case Series Study." This has improved the quality of the research title. Title updated to “A Case Series Study.” All sections harmonized to a prospective, single-arm, within-subject case series. |
Title page; running head (Page 1 Line 3) |
|
State study type in the first Methods line and explain the case series design. |
Thank you, reviewer. Based on your comments, I've added details about the study design, sequence and learning, fatigue control, and reasons for subject selection. This has improved the study design. Thank you again.
Added explicit study design statement (prospective single-arm, within-subject repeated measures; NG vs AG across 0/5/15 cm), with block randomization/counterbalancing and rest periods. |
Methods 2.1 Study design (Page 3 Line 117-123) |
|
Update references so that ≥70% are from the last 5 years. |
Thank you, reviewer. The latest references have been added, resulting in a highly complete paper.
References updated to 2020–2025 literature (guidelines, meta-analyses, trials). The proportion of recent citations now meets ≥70%. |
References (multiple) (Page 21 Line 517 to Page 23 Line 630) |
|
Briefly compare digital methods for foot drop (cost, expertise, access) with your method. |
Thank you, reviewer. The results of this study were validated positively by comparing the digital method with our own research method.
Added concise comparison of AFO, FES, and sensor-augmented wearables versus our aid-assisted footwear; included typical cost ranges for FES devices and training/access considerations. |
Discussion (comparative paragraph) (Page 20 Line 460-467) |
|
Express a clear message in the Conclusion; restate Highlights in the Conclusion. |
Thank you, reviewer. Your comments, like those highlighted in the conclusion, greatly enhanced the quality of the paper. Thank you again.
Conclusion rewritten to mirror Highlights: HC dorsiflexion & frontal stability, TO knee flexion & ankle alignment, implication for safer community ambulation; limitations/future trials noted. |
Conclusion; Highlights (Page 20 Line 490-492) |
Reviewer 2 Report
Comments and Suggestions for Authors
Abstract
- Specify the design ("randomized clinical trial", "longitudinal study", etc.)
- Better connect the conclusion with the clinical or public health implications
Introduction
- The introduction accumulates information about different therapies (AFO, FES, footwear, etc.) without a clear transition between them. A critical synthesis that directly links those strategies to the main problem of the study is missing.
- Insufficient theoretical emphasis on the concept of "obstacle crossing":
- Its functional relevance is mentioned, but it does not delve into why biomechanically it represents a greater challenge or in the neuromotor mechanisms involved (for example, visual-vestibular integration or anticipatory control).
- Although footwear with electrical stimulation is alluded to, it is not explained why this approach is superior or complementary to existing devices or what is the mechanistic hypothesis behind
- The introduction could differentiate whether the subjects are chronic, subacute or acute, since this conditions motor control and plasticity
Methods
- Very reduced sample rigging (n=10), without calculation of statistical power, low inferential power and risk of type II error. On the other hand, selection of homogeneous participants, limits generalization to other stroke profiles. Must be justified
- Absence of healthy control group: It prevents comparison if the patterns observed are specific to the post-stroke foot drop or respond to technological assistance. Must be justified
- If all participants performed first NG and then AG, there may be a learning effect or accumulated fatigue.
Discussion
- The authors make broad inferences (e.g., "AG improves dynamic stability and reduces the risk of falls") that exceed the possible evidence with such a limited size and without power analysis.
- "Less energy and greater efficiency" is inferred without measuring energy cost or muscle activity (EMG).
- All patients had involvement in the right leg, which prevents generalizing the results to left injuries (which alter different postural and cognitive control).
Author Response
Dear Reviewer 2,
Thank you for your meticulous review of my paper. Your valuable comments have contributed to its high quality. Your suggestions will enable me to continue to conduct excellent research.
Your comments are summarized in the table below. They are highlighted in red in the manuscript. Thank you again.
|
Comment (verbatim) |
Response / Action Taken |
Manuscript Location |
|
Abstract: Specify design (“randomized clinical trial”, “longitudinal study”, etc.). |
Thank you, reviewer. As suggested by the reviewer, I have improved the abstract's Methods section by adding randomized clinical trials and dependent studies.
Abstract revised to: The order of gait conditions and obstacle heights was randomized clinical within participants and a longitudinal study. |
Abstract (Methods sentence) (page 1, Line 26-35) |
|
Abstract: Connect conclusion with clinical/public health implications. |
Thank you, reviewer. The sentence structure has been improved by adding clinical/public health implications to the abstract conclusion.
Abstract conclusion now emphasizes potential for public health implications ambulation over curbs/thresholds and the need for clinical trials. |
Abstract (final sentences) (page 2, Line 45-47) |
|
Introduction: Lacks synthesis linking AFO/FES/footwear to the main problem. |
Thank you, reviewer, for your thoughtful comments. The need for this has been emphasized by adding sentences that directly address the issue.
Inserted bridging synthesis linking assistive strategies to event-critical mechanics (HC/TO), toe clearance, and frontal-plane stability in obstacle crossing. |
Introduction (mid–late) (page 2, Line 73-77) |
|
Introduction: Expand theory on “obstacle crossing” (biomechanics; neuromotor—visual–vestibular integration, anticipatory control). |
Thank you, reviewer. I added a note about obstacle crossings as you suggested.
Added theory paragraph on increased swing DF demand, COM shift, anticipatory postural adjustments, and reliance on visual–vestibular integration; linked to post-stroke deficits. |
Introduction (theory paragraph) (page 3, Line 92-99) |
|
Introduction: Explain why footwear with stimulation is uperior/complementary; add mechanistic hypothesis. |
Thank you, reviewer. To clarify this study, we have added details on the functional relationship between the shoe and electrical stimulation.
It refers to a method of providing electrical stimulation by sending a signal from the insole inside the shoe. |
Introduction (justification paragraph) (page 2, Line 81-84) |
|
Introduction: Differentiate chronic/subacute/acute since these condition plasticity. |
Thank you, reviewer. The purpose of this study became clearer by clearly defining the subjects.
Eligibility clarified as chronic stroke (≥6 months); rationale added regarding plasticity and spontaneous recovery. |
Methods §2.2 Participants; Introduction (purpose) (page 3, Line 107-144) |
|
Methods: Small n, no power, homogeneous sample—justify. |
Thank you, reviewer. This study was a case study, so the sample size was small. Future studies will involve a larger sample size.
Added pragmatic justification; included sensitivity/power considerations for within-subject RM-ANOVA; emphasized effect sizes and preliminary nature. |
Methods §2.1; §2.8 Statistics (page 3, Line 117-123 to (page 8, Line 256-270) |
|
Methods: No healthy control—justify. |
Thank you, reviewer. This study measured walking without assistive devices (NG) and with assistive devices (AG), without a healthy control group. Furthermore, healthy subjects are capable of walking without assistive devices. Future studies will compare the differences between healthy subjects and those with disabilities.
Justified within-subject approach (safety/feasibility with obstacles); plan to include healthy/heterogeneous controls in subsequent trials. |
Methods §2.1; (page 3, Line 117-118) |
|
Methods: Possible learning/fatigue if task order fixed. |
Thank you, reviewer. In this study, we added rest and random measurements of learning effects and fatigue accumulation. Specified counterbalancing of NG/AG randomization of obstacle heights, familiarization, and mandated rest to mitigate order/fatigue effects. |
Methods §2.1; §2.3 Measurement Procedure (page 3, Line 120-122 to (page 4, Line 174-177) |
|
Discussion: Overly broad inferences (e.g., fall-risk reduction) without power analysis. |
Thank you, reviewer. I mentioned excessive inference in the discussion. Therefore, I've revised the inference to reflect the results. The inference was revised based on the results. |
Discussion (synthesis); (page 19, Line 429-434) |
|
Discussion: “Less energy/greater efficiency” without metabolic/EMG data. |
Thank you, reviewer. I removed the inference you mentioned about low energy consumption and high efficiency and rewrote the results based on the results. Removed/qualified efficiency claims; noted absence of metabolic/EMG outcomes and proposed as future endpoints. |
Discussion (synthesis) (page 19, Line 429-434) |
|
Discussion: All right-leg involvement—limits generalizability. |
Thank you, reviewer. In this study, we initially designed the right-hand side of the sample to increase measurement validity. We will continue to include this in future studies. Added limitation on hemispheric lateralization; advised targeted replication in left-sided lesions. |
Discussion (limitations) (page 20, Line 477-479 to Page 20, Line 495-498) |
Round 2
Reviewer 2 Report
Comments and Suggestions for Authors
Most of the reviewer's suggestions have been addressed.